# The Effect of Environmental Regulation on the Efficiency of the Green Economy: The Intermediary Role of the Aggregation of Innovative Elements

**DOI:** 10.3390/ijerph20054575

**Published:** 2023-03-04

**Authors:** Xiaoyang Guo, Xiuwu Zhang

**Affiliations:** School of Statistics, Institute of Quantitative Economics, Huaqiao University, Xiamen 361021, China

**Keywords:** environmental regulation, agglomeration of innovation factors, green economic efficiency, super efficiency slacks-based measure model

## Abstract

Green development is the only way to realize harmonious coexistence between people and nature, so it is of great significance to create a benchmark for high-quality development. Based on the panel data of 30 provinces (except Tibet, Hong Kong, Macao, and Taiwan) in China from 2009 to 2020, the super-efficiency slacks-based measure model was used to calculate the green economic efficiency of various regions in China, and a related statistical model was used to verify the influence of different types of environmental regulation policies on green economic efficiency and the intermediary effect of innovation factor agglomeration. The results show that: (1) during the inspection period, the influence of public-participation environmental regulation on the efficiency of the green economy presents an “inverted U” trend, while command-control and market-incentive environmental regulation policies inhibit the improvement of green economic efficiency; (2) the agglomeration of innovative elements plays a significant intermediary role in the transmission path of environmental regulation affecting green economic efficiency, but the intermediary effects of different types of environmental regulation are slightly different. Finally, we discuss environmental regulation and innovative elements, and some corresponding suggestions are put forward.

## 1. Introduction and Literature Review

Since the implementation of the reform and opening-up policy, China has realized great achievements in terms of economic development, but at the cost of increasingly depleted resources and the deterioration of the environment. The report of the 20th National Congress of the Communist Party of China pointed out: “To focus on development in the new era, we must put more emphasis on the new development concept and accelerate the green transformation of the development model.” Admittedly, China has now reached an important period when it has to change the environmental model of “sacrificing the environment for economic growth.” Instead of blindly focusing on GDP growth, we should significantly improve the greening of the economy, and make resources and production and consumption factors compatible with each other. This is of great significance for comprehensively promoting the great rejuvenation of the Chinese nation with the Chinese path to modernization. In order to achieve sustainable economic development and alleviate the negative effects of environmental pollution, the “Fourteenth Five-Year Plan” clearly points out that it is necessary to establish and improve an environmental governance system that integrates above-ground, underground, land, and sea factors, and to intervene in the activities of market players through environmental regulations and other policy means, so as to guide the economy towards environmentally friendly, benign development. Therefore, we explore how different environmental governance tools affect the development of the green economy and whether they can effectively solve the “externality” problem to improve the efficiency of the green economy, which is of great significance for achieving a win-win situation between environmental protection and economic development and consolidating the foundation of high-quality economic development.

Generally, with strict compliance, environmental regulation tools can effectively convince market operators to adjust their internal resource allocation, improve the efficiency of green technology innovation, and reduce environmental regulation costs as far as possible, so as to increase their market share, alleviate local pollution control problems, and improve the level of regional green development, which conforms to the so-called “Porter hypothesis” of environmental regulation [1]. The Porter hypothesis states that appropriate environmental regulations can promote technological innovation of enterprises, improve the production efficiency of enterprises in the long term, thus offsetting the costs brought by environmental regulations, enhance the competitiveness of enterprises, and ultimately promote green economic growth. Therefore, whether strict environmental regulation can improve environmental quality and achieve economic growth at the same time or whether it leads to a decline in the efficiency of the green economy is still a controversial question in academic circles. On the one hand, scholars believe that environmental regulation is subject to the “innovation compensation” effect—that is, the policy itself promotes the transformation and upgrading of the industrial structure, makes the production factors flow to industries with lower pollution, eliminates backward production capacity, and at the same time, stimulates the green innovation consciousness of market players, improves the utilization efficiency of input factors in the production process, and creates a win-win situation in terms of environmental regulation and technological innovation [2,3,4,5]. On the other hand, environmental regulation tools have a “cost-regressive effect”. Due to the increase in energy saving, emission reduction, and pollution control costs, enterprises and other production entities need to adjust their scale of production, which will reduce their productivity and market competitiveness to some extent [6,7,8]. A few scholars believe that the positive promoting effect of environmental regulations on the development of the green economy is nonlinear, and the positive effect will only appear when the intensity of regulations reaches a certain level [9,10]. In addition, different types of environmental regulation tools have heterogeneous impacts on the green transformation and upgrading of regional industries and the improvement of the efficiency of the green economy. Moreover, the measures taken by market players to cope with heterogeneous environmental regulation tools are slightly different. Policy tools relying on administrative means and market regulation have a strong blocking effect on ecologically destructive behavior, while regulatory policies that only rely on public participation in supervision and control have room for improvement in terms of the efficiency of the green economy [11,12].

In summary, the existing literature analyzes the relationship between environmental regulations and green economic efficiency from different perspectives, providing ideas for how to study and analyze the relationship between environmental regulation, innovation factors, and green economic efficiency, but there is still marginal room for improvement. The contribution of this paper is as follows. First, the existing literature presents many contradictory conclusions as to whether environmental regulation tools are conducive to the improvement of green economic efficiency and the promotion of carbon emissions reduction, which leaves space for this paper to analyze the relationship between the two. Second, few studies have focused on whether the agglomeration of innovation factors has an incentive effect or a crowding-out effect on the development of the green economy. In view of this, we use a statistical model to quantify the impact of environmental regulations on the efficiency of the green economy, and explore the heterogeneity of innovation factor agglomeration under different types of environmental regulation tools.

The rest of this paper is organized as follows. Section 2 analyzes the impact of heterogeneous environmental regulation policies on the efficiency of the green economy and the role of innovation factor agglomeration in the transmission path, and puts forward the research hypothesis of this paper. Section 3 introduces the selection of variables, measurement models, and data sources. Section 4 gives the empirical results and analysis, including a benchmark regression analysis, robustness test, endogenous treatment, and the intermediary effect test of innovation factor agglomeration. Section 5 provides conclusions and policy recommendations, and puts forward the direction and content for further research in the future.

## 2. Mechanism Analysis and Research Hypothesis

### 2.1. Impact of Heterogeneous Environmental Regulation Policies

#### 2.1.1. Command-Controlled Environmental Regulation Policies

Command-controlled environmental regulation usually refers to environmental protection and administrative penalties formulated and promulgated by legislative or administrative departments, and mandatory provisions on the allowable content of pollutants [13]. Market operators can only passively accept and abide by the rules and policies formulated by the government. Different from market-oriented environmental regulation, the means of environmental regulation in China usually makes use of public power to force enterprises to make corresponding behavioral decisions, such as increasing the pollutant treatment level or adjusting the composition of production factors to control emissions and avoid punishment [14]. Mandatory measures have made some enterprises unable to adapt, or the cost of policy implementation is too high, so the relevant departments in these cases are unable to carry out long-term dynamic monitoring or adopt selective, superficial, and negative implementation of laws and regulations, which makes the final environmental regulation effect unsatisfactory and even leads to a reduction in efficiency of the green economy [15]. According to the 2021 Climate Change Performance Index, China’s climate change performance index is “medium” and the subsequent policy performance index is “high,” while the energy utilization and greenhouse gas emissions performance indexes are both “very low”. It can be seen that the implementation of environmental regulation in China is inconsistent between central and local governments, which has an adverse impact on the promotion of environmental improvement.

#### 2.1.2. Market-Incentivized Environmental Regulation Policies

Market-incentivized environmental regulation policies mainly guide enterprises and other business entities to incorporate negative externalities, such as environmental pollution caused by their production process, into their internal costs by collecting pollution charges and other market means, thus encouraging them to improve their production processes. Compared with command-controlled environmental regulation policies, enterprises and other market entities have more independent options and can flexibly adjust the factor structure according to their own production and operation conditions. The effective implementation of market-inspired environmental regulation often depends on a good market regulation mechanism. However, in the context of the inconsistency of environmental tax rates in different regions, it cannot promote technological innovation and industrial agglomeration, but is prone to breeding the phenomenon of “environmental rent-seeking” and increasing the burden of environmental governance for enterprises [16]. In addition, with the continuous development of China’s marketization process, the government’s participation in factor resource allocation is increasing, which leads to an imbalance in inter-regional market regulation, and even factor mismatch or price distortion.

#### 2.1.3. Public-Participation Environmental Regulation Policies

Public-participation environmental regulation policies mainly come from the pressure of individuals and social organizations on enterprises. With awareness of environmental protection growing among ordinary people, together with the popularity of real-time communication platforms such as TikTok, the openness and transparency of environmental information have been greatly enhanced, and the public can easily access environmental information, monitor the implementation of environmental policies, and express their views through online media. China’s environmental governance system adopts the multi-governance method of “government–enterprise–public,” which allows public opinion and public behavior to put pressure on enterprises’ behavior, forcing them to change environmentally destructive production processes or eliminate backward production capacity in order to avoid harsh administrative intervention [17]. In other words, when a pollution event occurs, stakeholders and the public can exert pressure on polluting enterprises through collective negotiation, media exposure, letters, and reports, and force enterprises to reduce their pollution emissions and increase green investment. According to stakeholder theory, in order to meet the public’s demand for environmental protection, enterprises will continuously adjust their own behavior to ensure the reasonable and stable allocation of resources of various production factors. At the same time, due to the consideration of maintaining their own image and social reputation, production and operation entities will overcome the short-sighted behavior of pursuing current interests, increase research and development into green technologies, or commit funds to realize the transformation from “terminal pollution control” to “internal and external control,” and occupy a dominant position in the market.

### 2.2. The Mediating Effect of Innovation Factor Agglomeration

The agglomeration of innovation factors is a dynamic process of continuous accumulation, allocation, and integration [18]. As China pays more attention to environmental protection, relevant administrative regulations have been issued to force all kinds of market players, especially enterprises with high pollution and high emissions, to employ more green technology innovation so as to eliminate negative externalities, establish a good corporate image, and occupy a greater market share. Specifically, according to the Porter hypothesis, environmental regulation policies can encourage enterprises to improve their production processes and pollution control technologies spontaneously to a certain extent. Communication and cooperation with enterprises about advanced green production technologies will result in the continuous flow, interaction, and optimization of innovation factors, and produce a certain innovation compensation effect, which may even exceed the environmental regulation costs borne by enterprises and realize the two-way coordination of economic and ecological benefits. At the same time, the “strictest-ever” environmental protection system, implemented after the 18th CPC National Congress, has sent a strong signal to market operators and the public. The demand for environmental protection products in the end consumer market has increased, resulting in a market preference for green products and services. This also indicates that the economic benefits generated by environmental regulation not only come from production frontier displacement, but also from the long-term, hidden benefits brought about by the regional agglomeration of innovation factors, such as organizational management experience, human capital, and related technical measures.

### 2.3. Research Hypothesis

Based on the above theoretical analysis, we propose the following research assumptions.

**Hypothesis 1.** 
*Command-controlled environmental regulation policies have a restraining effect on the improvement of green economic efficiency due to their mandatory nature and high implementation cost.*


**Hypothesis 2.** 
*Market-incentivized environmental regulation policies have a restraining effect on the improvement of green economic efficiency due to the inconsistency of environmental tax rates between regions and the excessive participation of the government in resource allocation.*


**Hypothesis 3.** 
*Due to the popularity of online media and the public’s supervision of the production behavior of enterprises, public-participation environmental regulation policy can promote the efficiency of the green economy.*


**Hypothesis 4.** 
*The agglomeration of innovation factors generated by environmental regulation policies will bring long-term economic benefits to the market and help promote the improvement of green economic efficiency.*


## 3. Research Design

### 3.1. Variable Selection

#### 3.1.1. Explained Variable: Green Economy Efficiency (GEE)

The super-efficiency slacks-based measure (SBM) model proposed by Tone (2003) combines the advantages of the super-efficiency data envelopment analysis (DEA) model and the SBM model [19]. While dealing with unexpected output, it compares effective decision-making units, makes up for defects in the traditional efficiency model, and solves the problem of it being difficult to measure green economic efficiency due to the constraints of resources and the environment. The higher the efficiency value of the green economy, the less pollution is caused by the output of the limited resource input.

In view of this, the SBM model considering unexpected output is as follows:$$\rho^{*}=\min\frac{1-\frac{\sum_{i=1}^{m}\frac{s_{1}^{-}}{x_{i0}}}{m}}{1+\frac{1}{s_{1}+s_{2}}(\sum_{r=1}^{s_{1}}\frac{s_{r}^{g}}{y_{r0}^{g}}+\sum_{r=1}^{s_{2}}\frac{s_{r}^{b}}{y_{r0}^{b}})}$$
$$s.t.\;\left\{\begin{array}{c} x_{0}=X\lambda +s^{-}\\ y_{0}^{g}=Y^{g}\lambda -s^{g}\\ y_{0}^{b}=Y^{b}\lambda +s^{b} \end{array}\right.$$

At the same time, $s^{-}\geq 0,s^{g}\geq 0,s^{b}\geq 0,\lambda \geq 0$, where s represents the relaxation variable of input and output, λ is the weight, and $\rho^{*}$ is a strictly monotonically decreasing function of $s^{-},\;s^{g},\;s^{b}$ whose value is between 0 and 1. In addition, this paper refers to the practice of Qian (2013) to transform the nonlinear model into a linear model [20], as follows:$$\tau^{*}=\min(t-\frac{\sum_{i=1}^{m}\frac{s_{i}^{-}}{x_{i0}}}{m})$$
$$s.t.\;\left\{\begin{array}{c} 1=t+\frac{1}{s_{1}+s_{2}}(\sum_{r=1}^{s_{1}}\frac{s_{r}^{g}}{y_{r0}^{g}}+\sum_{r=1}^{s_{2}}\frac{s_{r}^{b}}{y_{r0}^{b}})\\ x_{0}t=X\lambda +S^{-}\\ y_{0}^{g}t=Y^{g}\lambda -S^{g}\\ y_{0}^{b}t=Y^{b}\lambda +S^{b} \end{array}\right.$$

This satisfies $s^{-}\geq 0,s^{g}\geq 0,s^{b}\geq 0,\lambda \geq 0,t>0$.

When measuring the efficiency of the green economy through the super-efficiency SBM model, it is necessary to set the expected output and unexpected output of the development of the green economy. Therefore, we referred to Lin [21] (2019), Li [22] (2022), and Wang [23] (2022) to build a green economy efficiency evaluation system, including factor input indicators, expected output, and unexpected output indicators.
(1)Factor input. Considering the differences between provinces in terms of economic development level, price index of investment goods, and depreciation rate, as well as the availability of data, we used annual fixed asset investment to represent capital input (K). For labor input (L), we used the actual total number of employees at the end of the year (Wang [24], 2020). As for energy input (EI), the academic community has not reached a consensus on energy input index; some scholars use the energy consumption per unit of GDP to represent it (Liu [25], 2019). Through index combining and comparison, we found we can avoid the errors caused by the differences in energy structure between regions to a certain extent. Therefore, we chose the total energy consumption of regions as representative in this paper.(2)Expected output. In order to truly reflect the economic operation status of each province, we chose real GDP as the expected output variable, and calculated the real GDP based on 2009 values.(3)Unexpected output. In this paper, the major environmental pollution sources in the process of economic and social development were selected as the index of unexpected output, including the wastewater, waste gas, and solid waste pollution caused by industrial enterprises’ production, which are measured by industrial wastewater emissions, industrial sulfur dioxide emissions, and industrial smoke and dust emissions, respectively.

#### 3.1.2. Core Explanatory Variable: Environmental Regulation (E)

The existing literature has not set the metric of environmental regulation uniformly. This paper refers to the practices of Gao [26] (2015), Zhang [27] (2015), and Peng [28] (2019). The following indicators are selected to represent environmental regulations: command-controlled environmental regulations, mainly reflected in the emissions and disposal of pollutants carried out by enterprises according to regulatory standards. The proportion of the total environmental investment in the construction of three simultaneous projects in the total industrial added value is selected as representative. Market-incentivized environmental regulation is mainly reflected in increasing the external costs of enterprises by means of tax or pollutant emissions permit trading and other tools to encourage them to reduce the level of pollution discharge. The pollutant discharge fees levied by provinces are used as representative. Public-participation environmental regulation is mainly reflected in the external pressure exerted on enterprises by individuals and social organizations involved in the supervision of enterprises’ pollutant discharge behavior. It is represented by the sum of the total number of environmental letters and visits of each province.

#### 3.1.3. Intermediary Variable: Innovation Factor Agglomeration (IE)

Based on the China Innovation Index (CII), we constructed an innovation factor agglomeration evaluation system from the perspectives of innovation input and output, and used the entropy method to calculate the innovation factor agglomeration index. Specifically, the index of innovation investment was measured by the index of R&D expenditure and R&D personnel investment, which reflects the scale of scientific research talent and the exploration intensity of innovation ability in each region to a certain extent. The index of innovation output is measured by the number of effective invention patents and the transaction amounts of the technology market, which better reflects the technology R&D capacity and the transaction scale of the research market in each region.

#### 3.1.4. Control Variables

In order to ensure the reliability and objectivity of the research results, we selected the following control variables, considering the influencing factors of the efficiency level of the green economy. (1) Industrial structure. Considering that the proportion of environmental effects and green economic effects contributed by tertiary industry in the process of social and economic development is relatively large, we should refer to the practice of Gan [29] (2011) and measure it as the proportion of the added value of tertiary industry to the added value of secondary industry. (2) Infrastructure construction. A good infrastructure provides a convenient social environment for the smooth development of economic activities, thus reducing the operating costs of enterprises and enhancing their ability to attract foreign investment. Therefore, according to the practice of Wang [30] (2021), we selected the per-capita road area and carried out logarithmic processing. (3) Degree of opening-up. The introduction of foreign advanced factor resources can accelerate domestic technological reform and institutional innovation, thus improving economic production efficiency. However, at the same time, the introduction of high-pollution and high-emissions enterprises has increased the pressure on the local environment. Therefore, with reference to the practice of Dong [31] (2021), we selected the proportion of the actual use of foreign investment to the total GDP. (4) City size. The size of a city is positively correlated with the service category and scope of the region. However, with the deepening of the urbanization process, the sudden rise in “urban diseases” such as personnel congestion and resource waste hinder the development of the urban green economy. Therefore, with reference to Wu [32] (2021), we selected the urban population density at the end of the year to represent this. (5) Energy consumption. Considering that irrational energy consumption leads to excessive waste of resources and a low utilization rate, which leads to serious environmental pollution, we followed Zhang [33] (2022) in using the total energy consumption at the end of the year as representative. The descriptive statistics of the above selected variables are shown in Table 1.

### 3.2. Econometric Model

The core purpose of this paper is to verify the impact of environmental regulation on green economic efficiency and identify the intermediary effect of innovation factor agglomeration in its transmission process. In order to verify the hypotheses and mechanism of action above, we used a statistics model. Based on the above definition and the core explanatory variables and control variables, the following benchmark econometric model was constructed:$$lnGEE_{it}=\alpha_{0}+\alpha_{1}lnE_{it}+\alpha_{2}lnISS_{it}+\alpha_{3}lnPRA_{it}+\alpha_{4}lnOG_{it}+\alpha_{5}lnCIS_{it}+\alpha_{6}lnECS_{it}+\nu_{i}+\mu_{t}+\varepsilon_{it}$$
where α represents the parameter to be estimated; subscripts i and t represent the province and year, respectively; $\nu_{i}$ represents the individual fixed effect; $\mu_{t}$ represents the time fixed effect; and $\varepsilon_{it}$ represents the random disturbance term subject to white noise. In the actual fitting calculation process, in order to mitigate the influence of heteroscedasticity and reduce the data level of variables, logarithmic transformation of all variables was carried out. In addition, in order to verify the intermediary effect of the agglomeration of innovative elements in the transmission process of environmental regulation affecting the efficiency of the green economy, following the theoretical ideas provided by Wen [34] (2004), recursive equations were used for testing, and then the following equations were constructed:$$lnGEE_{it}=c_{1}+\beta_{1}lnE_{it}+\overset{n}{\underset{j=1}{\sum }}\gamma_{j}\;lnControl_{jit}+\varepsilon_{it}$$
$$lnIE_{it}=c_{2}+\beta_{2}lnE_{it}+\overset{n}{\underset{j=1}{\sum }}\gamma_{j}\;lnControl_{jit}+\varepsilon_{it}$$
$$lnGEE_{it}=c+\beta_{1}lnE_{it}+\beta_{2}lnIE_{it}+\overset{n}{\underset{j=1}{\sum }}\gamma_{j}\;lnControl_{jit}+\varepsilon_{it}$$

Considering that stepwise regression has low testing efficacy, we used the Sobel test to explore the intermediary effect of innovation factor agglomeration, and it mapped the mediating effect well [35].

### 3.3. Data Sources

According to the principle of data availability, panel data of 30 provinces in China (except Xizang, Hong Kong, Macao and Taiwan) from 2009 to 2020 were selected as research samples. The original data of all variables were mainly from the China Environmental Yearbook, China Environmental Statistical Yearbook, the National Intellectual Property Office, the National Bureau of Statistics, the EPS database, the China Economic Network database, and provincial and municipal statistical yearbooks. For the very few missing values, the Lagrange interpolation method was used.

## 4. Empirical Testing and Analysis of Results

### 4.1. Analysis of Benchmark Regression Results

Common statistical models used for panel data include the pooled ordinary least square method (POLS), random effects model (RE), and fixed effects model (FE). Which specific method is most suitable for the sample data in this paper needs further examination. The test results showed that both the F-test and Hausman test were significant at the 1% significance level, leading us to reject the null hypothesis and indicating that the FE model is optimal. In view of this, we selected the fixed effect model as the benchmark regression model for subsequent empirical tests. Meanwhile, in order to eliminate the interference of heteroscedasticity, sequence correlation, and cross-correlation on regression results, we mainly considered Driscoll–Kraay standard error. In addition, to determine whether there is a nonlinear relationship between different types of environmental regulation tools and green economic efficiency, the square terms of each environmental regulation variable were included in the model. STATA 16.0 software was used for fitting the calculation according to the equation set above, and the following results were obtained.

As can be seen from Table 2, different types of environmental regulation tools all have a certain impact on the efficiency of the green economy, which is at least significant at the significance level of 5%.

Specifically, with control of both time effect and year effect, the average estimation coefficient of command-controlled environmental regulation on green economic efficiency is −0.122 and passes the 1% significance test, indicating that command-controlled environmental regulation tools are not conducive to the green development of the Chinese economy. The reasons may lie in the following aspects. First, some laws and regulations fail to accurately identify the types of enterprises, and force enterprises to undertake corresponding responses by means of public power. There is a “one-size-fits-all” attitude, which leads to unsatisfactory environmental regulation. Second, there is a time lag in the transformation of the development mode of the green economy. Due to the lack of innovation impetus, some enterprises can only reduce the production scale or increase the environmental treatment fees to control the emissions of pollutants in order to achieve the established policy and regulation goals; they are prone to resisting mandatory measures and do not want to sacrifice their own production efficiency, resulting in the “cost compliance” effect.

As can be seen from the second column of the table, the average estimated coefficient of market-incentive environmental regulation on the efficiency of the green economy was −0.278, which passes the 1% significance test, indicating that market-incentive environmental regulation tools are also unfavorable to the development of the green economy. This conclusion may be related to the imperfect market regulation mechanism and the immature carbon emissions trading permit market in China. Li (2020) proposed that the marketization process can improve the effectiveness of environmental regulation tools and is closely related to the Porter effect [36]. At the same time, the internalization of enterprise costs by market-incentive environmental regulation tools leads to a substantial increase in operational and management costs, which naturally reduces funds for the research and development of green products and green technology, which is not conducive to the transformation and upgrading of enterprises.

The average estimated coefficient of public-participation environmental regulation on green economic efficiency was 0.069, which passes the 5% significance level test. This indicates that public-participation environmental regulation tools can promote the development of the green economy. As the public pays more and more attention to environmental issues, they are more actively participating in the supervision of enterprises’ pollutant discharge behavior, and enterprises, limited by the consideration of maintaining their own image and social reputation, will optimize their own production behavior in advance, accelerate green technology innovation to avoid strict administrative intervention, and realize the transformation from “end pollution control” to “source pollution control”.

In addition, the fitting coefficients of the square term of command-control environmental regulation and market-incentive environmental regulation were −0.002 and −0.013, respectively, which did not pass the significance test, while the average estimated coefficient of the square term of public-participation environmental regulation on green economic efficiency was −0.029 and was significant at the 1% level. This indicates that there is an “inverted U-shaped” relationship between public-participation environmental regulation and green economic development.

### 4.2. Robustness Test

The results in Table 2 reveal the basic relationship between different types of environmental regulation tools and the efficiency of the green economy. In order to verify the robustness and reliability of the baseline regression estimation results, the following three methods were adopted for demonstration and illustration.

(1) Shrink-tail treatment. It prevents the deviation of the regression results caused by outliers: for example, during the COVID-19 pandemic, all of China’s industries were affected by economic shocks; or there were major natural disasters; or, because of major technological breakthroughs in a certain region, green technologies were promoted and applied rapidly. Therefore, with reference to Sun [37] (2020), all the continuous variables were reduced by 1% and then re-estimated by the FE model. The regression results are shown in columns (1) to (3) in Table 3.

(2) Change the explained variable. Considering that the measurement methods of expected output and unexpected output have not been unified in the process of measuring the efficiency of the green economy at present, there is deviation in the final measurement results. In this paper, the unexpected output index of chemical oxygen demand (COD) was added on the basis of the original sulfur dioxide emissions to calculate the new green economic efficiency level and conduct another regression test. The estimated results are shown in columns (4) to (6) in Table 3.

(3) Replace the model. When the random disturbance term has intragroup correlation, intergroup correlation, and concurrent correlation, the estimation results of the bidirectional fixed-effect model may have some bias. In addition, Driscoll−Kraay standard error was used to solve the possible heteroscedasticity, autocorrelation, and cross-correlation problems in the above reference regression analysis. However, in addition to using this standard error, feasible generalized least squares (FGLS) can also deal with three major threats to short panel data. Due to the small number of cross-sections, we allowed each individual to have the same autoregressive coefficient in the estimation process, and used the AR (1) autocorrelation structure unique to the panel data. The estimation results are shown in columns (7) to (9) in Table 3.

It is not difficult to see that the impact of different types of environmental regulation tools on green economic efficiency is roughly the same as the benchmark regression results; at least the positive and negative and significance of the fitting coefficient has not changed significantly, so the benchmark regression results can be considered reliable and robust.

### 4.3. Endogenic Processing

Usually, endogenous problems involve missing important variables, measurement deviation, and mutual causality. Although, considering the relationship between environmental regulation and green economic efficiency, we sought to alleviate the endogenous problems caused by missing variables and measurement errors by selecting more control variables, the model setting still faces the threat of mutual causal endogenous problems between environmental regulation tools and green economic efficiency. In view of this, the two-stage least square method (2SLS) was adopted. With reference to the practices of Hering [38] (2014) and Shen [39] (2017), the air circulation coefficient (IV) was selected as a tool variable. When the air circulation coefficient is low, pollutants such as PM2.5 and smoke dust emitted by manufacturing and industrial industries are suspended in the atmosphere, and it is difficult to remove them, thus increasing the environmental regulation efforts of the local government. In this paper, the data of wind speed and atmospheric boundary layer height at 10 m published by the ERA database of the European Center for Medium-Range Weather Forecasts (ECMWF) were used to calculate the air circulation coefficient from 2009 to 2020 using ArcGIS10.5 software, and the circulation coefficient was matched according to the longitude and latitude information of provincial capitals. The construction method was as follows:(1)$$AC_{it}=WS_{it}+BLH_{it}$$
where *WS* and *BLH* represent the wind speed and atmospheric boundary layer height at 10 m, respectively, and *AC* represents the air flow coefficient. In order to avoid the influence of weak instrumental variables, we selected the environmental regulation tool lag term as the second instrumental variable.

As can be seen from Table 4, the 2SLS method has passed the unrecognizable test, the weak instrumental variable test, and the over-recognition test, and it is considered that the instrumental variable is exogenous and irrelevant to the disturbance term. From the estimation results, the estimation coefficients of green economic efficiency of different types of environmental regulation tools are roughly the same, the positive and negative values remain consistent, and the significance changes slightly but does not affect the reliability of the research conclusions. Therefore, the effect of environmental regulation tools on the efficiency of the green economy supports the conclusion in the benchmark regression analysis after the elimination of possible endogenous problems.

### 4.4. Testing the Mediating Effect of Innovation Factor Agglomeration

In order to reveal whether innovation factor agglomeration plays a significant mediating effect in the transmission mechanism of environmental regulation’s impact on green economic efficiency, we used Sobel’s mediation effect test to perform fitting calculations on panel data according to the equation established above, and the specific results are shown in Table 5.

It can be seen from the table that the Z-values of the Sobel test of heterogeneous environmental regulations are all significant above 5%, which indicates that the mediating effect of innovation factor agglomeration is valid. The only difference is that command-and-control environmental regulation tools have slightly different transmission mechanisms. Specifically, the Sobel test values of market-incentive environmental regulation and public-participation environmental regulation are both positive and significant at the 1% significance level, indicating that there is a positive mediating effect. However, the Sobel value of command-controlled environmental regulation tools is significantly negative, which indicates that environmental regulation inhibits the growth in efficiency of the green economy through the intermediary path of innovation factor agglomeration. A possible reason behind this is that the current market operation mechanism is not mature, and so there are market distortion and factor mismatches to a certain extent. As a result, production enterprises under the restriction of administrative measures face high “menu costs” when entering and exiting the industry. Enterprises with low production efficiency and high pollution automatically cede their market shares. However, more efficient potential entrants are also discouraged by market entry barriers and policy uncertainties, which is not conducive to the geographical agglomeration of high-tech enterprises or enterprises with strong innovation vitality. In addition, enterprises in the industry are restricted by public power and do not have the corresponding financial and material resources to carry out green technology innovation in the short term, so the positive effect of innovation factor agglomeration is not activated.

## 5. Conclusions

Green economic efficiency is not only a key indicator to describe the green development of industries, but also an important indicator to measure the high-quality development of industries. We first used the super-efficiency SBM model to measure the green economic efficiency level of 30 provinces in China from 2009 to 2020, and then a series of mathematical statistical models to empirically test the impact of three different types of environmental regulation tools and innovation factor agglomeration on green economic efficiency. The results show that, first, heterogeneous environmental regulations have slightly different effects on green economic efficiency. Among them, command-controlled environmental regulation and market-incentivized environmental regulation have an inhibitory effect on the improvement of green economic efficiency, while the impact of public-participation environmental regulation on green economic efficiency shows an “inverted U-shaped” relationship, which first promotes and then inhibits the improvement of green economic efficiency. Second, under the influence of the mediating factors of innovation factor agglomeration, public-participation and market-incentivized environmental regulation tools have a positive mediating effect on green economic efficiency, promoting the green development of Chinese industries. However, command-controlled environmental regulation tools cannot fully activate the positive utility of innovation factor agglomeration—that is, they inhibit the improvement of green economic efficiency.

The research conclusions of this paper provide an empirical basis for promoting the green development of China’s industry, improving the efficiency of the green economy, and achieving high-quality development. Therefore, the following policy recommendations are proposed:(1)Optimize the design of environmental regulation policies and improve the precision of environmental governance. Different types of environmental regulation policies have slightly different effects due to differences in implementation based on the regional economic development level, regional culture, and other factors. Therefore, there is no optimal environmental regulation policy to promote the development of the green economy nationally. The government should further promote the transformation of environmental regulation policy from relying on administrative means to comprehensively considering laws, fiscal policies, taxation, technology, and necessary administrative measures, and strengthen the application of compound environmental regulation tools. To be specific, after optimizing and improving top-level design and formulating environmental regulation policies according to local conditions, there should be no inefficient environmental protection behaviors that hinder the normal production and operation activities of enterprises and inhibit the production enthusiasm of enterprises under the pretext of strengthening industrial pollution control. At the same time, a performance appraisal system with green GDP as the core should be built to avoid government departments’ excessive pursuit of “nominal achievements” and the introduction of unsuitable enterprises without screening, so as to truly curb “GDP worship” and fundamentally solve the problems of ecological compensation and environmental pollution between regions. In addition, it is necessary to further improve the market-oriented mechanism of environmental regulation, price formation mechanism, and enterprise’ main rights and responsibilities allocation mechanism, safeguard the enterprise’s production management rights, and ensure that the benefits created by green technology innovation are reflected in production and business activities, to minimize the loss of administrative environmental governance efficiency.(2)Broaden the channels for public participation in environmental governance and bring into play the synergistic effect of environmental regulation tools. In order to give full play to the role of public participation in environmental regulation and governance, relevant functional departments should strictly implement the government information disclosure system; use WeChat official accounts, video accounts, and other media platforms to popularize and publicize environmental protection knowledge; and raise general awareness of environmental protection. At the same time, it is important to establish an effective “government−market−public” linkage mechanism, set up channels for public supervision and reporting, ensure that problems reported by the public receive timely feedback and handling, and create a social trend of public participation and supervision. It is worth noting that the irrational, non-objective, and non-legal pressure of public opinion in the implementation of environmental regulations should be avoided to prevent the “track overlap” of the impact of environmental regulations on market operators, resulting in the “green paradox” effect.(3)Give full play to the spatial spillover effect of innovation factor agglomeration, and accelerate green innovation and achievement transformation. Due to the forced effect of environmental regulation policies, enterprises are encouraged to speed up green technology innovation, which leads to a continuous flow of innovation elements in the region and the formation of an agglomeration phenomenon, resulting in new effective connection points and the improvement of overall production efficiency. Specifically, the government should increase investment in capital elements, improve the concentration level of R&D funds, provide market operators with appropriate subsidies and tax incentives for innovation, and give full play to the leader role of enterprises integrating specialization, refinement, and novelty through publicity of such “star” enterprises and the telling of green development stories, so as to stimulate the endogenous driving force of enterprise innovation. At the same time, undertaking industrial transfer in technology-intensive areas actively promotes the integration of digital intelligence into green development, updates cutting-edge production equipment to reduce energy consumption rate, scientifically controls pollution emissions in production links, and improves the overall technical efficiency level and resource utilization rate in the region.

Against the background of global carbon emission reduction and green economic transformation, this study provides quantitative evidence of the impact of different types of environmental regulation policies on the efficiency of the green economy, and explores whether environmental regulation policies can promote the efficiency of the green economy from the perspective of innovation factor agglomeration. This paper enriches knowledge of the mechanisms of environmental regulation from previous literature from both theoretical and empirical perspectives, and has practical value for achieving high-quality green development. Finally, it is worth noting that, although the purpose of this paper was to study the green effect of China’s environmental regulation policies, the proposed mechanism and research methods can be applied to study the impact of other countries’ environmental regulation measures on green economic efficiency.

Although this paper provides some enlightenment for the government’s decision-making and research in the field of environmental regulation and green economic growth, it still has certain limitations. First of all, due to the availability of data, we used provincial data to discuss the impact of environmental regulation policies on the efficiency of the green economy from 2009 to 2020. Future research can expand on the impact of environmental regulation policies in terms of timeliness by adjusting research methods and perspectives. Secondly, from the perspective of research objects, the enterprise level is not included in the research framework of this paper. The research results may show that enterprises have different reactions to different environmental regulation policies, and thus have different impacts on the efficiency of the green economy. Therefore, we need to study the micro-effects at the enterprise level to draw broad and profound conclusions.

## Figures and Tables

**Table 1 ijerph-20-04575-t001:** Definitions of variables.

	Variable Name	Abbreviation	Observations	Mean	Standard Deviation	Min	Max
Explained variable	Green economy efficiency	GEE	360	0.7031	0.2382	0.5197	1.0997
Factor input	Capital input	K	360	10.1372	10.3183	6.7811	11.9824
	Labor input	L	360	6.1359	5.7192	3.7496	7.5893
	Energy input	EI	360	9.2471	8.9268	5.9221	10.6110
Expected output	Real GDP	G	360	8.0437	7.7523	5.3089	9.3588
Unexpected output	Industrial waste water	WW	360	12.2004	12.0799	8.6225	13.7517
	Industrial waste gas	WG	360	4.1179	3.6904	0.3556	5.2081
	Solid waste	S	360	9.2159	9.0840	5.3025	10.3037
Explanatory variable	Command-controlled environmental regulation	AE	360	9.6148	2.9845	0.9761	15.4229
	Market-incentivized environmental regulation	ME	360	4.7331	2.5716	0.8906	17.8673
	Public-participation environmental regulation	PE	360	3.2146	1.0499	1.0276	3.4977
Intermediary variable	Innovation factor agglomeration	IE	360	7.9142	1.0936	3.6969	9.7201
Control variable	Industrial structure	ISS	360	0.9835	0.4759	0.4971	4.1653
	Infrastructure construction	PRA	360	4.7103	4.0579	2.3461	6.2925
	Degree of opening-up	OG	360	9.4332	8.5976	5.1850	12.1513
	City zize	CIS	360	0.6531	0.3308	0.4985	0.8966
	Energy consumption	ECS	360	9.2471	8.9268	5.9221	10.6110

**Table 2 ijerph-20-04575-t002:** Baseline regression analysis.

Variable	(1)	(2)	(3)	(4)	(5)	(6)
AE	−0.122 *** (–4.28)			−0.123 *** (–4.27)		
AE^2^				−0.002 (–0.34)		
ME		−0.278 *** (–5.60)			−0.281 *** (−4.56)	
ME^2^					−0.013 (–0.59)	
PE			0.069 ** (2.41)			0.053 * (1.84)
PE^2^						−0.029 *** (–2.85)
ISS	0.502 *** (2.59)	0.513 *** (2.58)	0.497 ** (2.43)	0.519 *** (2.40)	0.501 *** (2.12)	0.506 ** (2.27)
PRA	0.164 *** (5.48)	0.131 *** (4.47)	0.158 *** (5.45)	0.154 *** (5.94)	0.243 *** (6.22)	0.159 *** (5.45)
OG	0.078 *** (4.95)	0.077 *** (4.73)	0.084 *** (2.55)	0.091 *** (5.13)	0.079 *** (4.80)	0.085 *** (2.56)
CIS	−0.174 *** (–4.36)	−0.184 *** (–4.75)	−0.185 *** (–5.29)	−0.152 *** (–4.28)	−0.176 *** (–5.28)	−0.182 *** (–5.31)
ECS	−0.184 *** (–5.27)	−0.198 *** (–5.41)	−0.165 *** (–4.88)	−0.179 *** (–5.09)	−0.191 *** (–5.45)	−0.161 *** (–4.91)
Term of constant	−4.152 *** (–6.46)	−4.374 *** (–6.87)	−4.226 *** (–6.29)	−4.150 *** (–6.45)	−4.544 *** (–5.53)	−3.214 *** (–4.13)
Effect of individual	Control	Control	Control	Control	Control	Control
Effect of year	Control	Control	Control	Control	Control	Control
*R* ^2^	0.7188	0.7300	0.7146	0.7188	0.7356	0.7199

Note: ***, **, and * are significant at the level of 1%, 5%, and 10%, respectively. The *t*-statistic is reported in brackets.

**Table 3 ijerph-20-04575-t003:** Robustness test.

Variable	(1)	(2)	(3)	(4)	(5)	(6)	(7)	(8)	(9)
AE	−0.198 *** (−3.41)			−0.106 *** (−4.02)			−0.143 *** (−4.11)		
ME		−0.294 *** (−5.18)			−0.269 *** (−5.21)			−0.285 *** (−5.09)	
PE			0.149 ** (3.95)			0.072 ** (2.84)			0.109 ** (3.37)
Variable of control	Control	Control	Control	Control	Control	Control	Control	Control	Control
*R* ^2^	0.7486	0.7429	0.7380	0.7581	0.7598	0.7521	0.7124	0.7203	0.7199

Note: *** and ** are significant at the level of 1% and 5%, respectively. The *t*-statistic is reported in brackets.

**Table 4 ijerph-20-04575-t004:** Endogenic processing.

Variable	(1)	(2)	(3)
AE	−0.109 * (–4.11)		
ME		−0.264 *** (–5.29)	
PE			0.074 *** (2.49)
Variable of control	Control	Control	Control
Effect of individual	Control	Control	Control
Utility of years	Control	Control	Control
Unidentifiable test	169.947 ***	81.534 ***	116.451 ***
Weak instrumental variable checking	122.529	59.193	80.911
Overidentification test	1.732	2.049	2.583

Note: *** and * are significant at the level of 1% and 10%, respectively. The Z-statistic is reported in brackets.

**Table 5 ijerph-20-04575-t005:** Test of mediating effect.

Variable	(1)	(2)	(3)
AE	−0.207 *** (–3.98)		
ME		0.098 *** (2.67)	
PE			0.071 *** (3.14)
Sobel	−0.041 ** (–2.39)	0.093 *** (2.81)	0.061 *** (3.58)
Effect of mediation	0.1153	0.4952	0.1404
Variable of control	Control	Control	Control

Note: *** and ** are significant at the level of 1% and 5%, respectively. The Z-statistic is reported in brackets.

## Data Availability

Readers can obtain the raw datasets used in this paper by themselves through the data sources described in Section 3 or by contacting the first author or the corresponding author.

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
