# Peer review of "The Effect of Environmental Regulation on the Efficiency of the Green Economy: The Intermediary Role of the Aggregation of Innovative Elements"

_ijerph, 2023, doi:10.3390/ijerph20054575_

Round 1

Reviewer 1 Report

Thank you for the opportunity to review this interesting paper. Whether and how environmental regulation can be a source of economic efficiency, technological progress, and environmental improvement is an ongoing issue in the academic literature and among policy-makers.

The modeling of the Green Economy Efficiency as the explained variable as well as the explanatory variables is consistent and captures the major factors required. The regression and statistical analyses are appropriate for the task as is the interpretation. I cannot comment on the actual analytical processes because supplemental material was not provided. The three main conclusions/recommendations follow logically from the results.

There are several weaknesses that must be addressed in the revision:

An English copy-edit is essential. In many places, the authors' meaning is difficult to understand. I am sufficiently familiar with the literature and methods to decipher the text; however, readers of this journal with less familiarity will not be able to. (A few examples are included in the attached file.)

A more elaborate discussion of the results must be included. As the authors point out, the issue of the influence environmental regulation on performance is controversial and there is a great deal of literature with differing results. For this reason, the literature review, results and conclusion is insufficient. The agreement/disagreement of the authors' results with other literature must also be presented. 

Specific comments:  

Please make sure that the full name of all acronyms are used at the first appearance of the acronym.

The paper draws on seminal works in this field; however, insufficient explanation is given to explaining which aspects they emphasize. For example, it is not enough to mention the Porter hypothesis without stating it clearly. A reasonably good example of how this can be accomplished is in line 59-61 in which the innovation compensation is explained. This should be followed throughout (e.g., SBM technique, etc). 

There are multiple instances in which citations are incomplete (e.g., line 194, 207 , and 215 only the date appears).

INTRODUCTION: The introduction is rather difficult to follow. Although the literature is summarized, and the rationale for the approach taken is explained, the presentation of the literature itself is a bit jumbled. It would be preferable if the authors clearly distinguish the important theoretical aspects (e.g., Porter hypothesis, innovation effect, agglomeration factors, etc.)

Section 2: This section should be titled Methods.

Lines102-107 This is not a full sentence and its meaning is unclear. It seems important, but the point must be clarified.

Add counter hypothesis to each hypothesis

Section 2.1.3 The role of public participation in promoting or restricting "green economic efficiency" needs to be clarified. H3 is too broadly stated. Please clarify what aspects of policy or regulation the public is participating in. For example, is the public involved in policy-making or regulation setting or monitoring and enforcement, or all of the above?

LIne 122 Is signals the correct term? Incentives and disincentives seems more appropriate.

Line 27: Please explain the reform to which you are referring. Not all readers will be familiar with it.

Generally, check punctuation and spacing including one space between sentences. Numerous sentences have no spaces in between them.

Author Response

Please refer to the attachment "Reply to the review report".

Reviewer 2 Report

Based on panel data of 30 provinces in China (excluding Tibet, Hong Kong, Macau, and Taiwan) from 2009 to 2020, this article uses the super-efficient SBM model to perform the green economic efficiency of various regions in China and related mathematical calculations. A statistical model was used to validate the impact of different types of environmental regulation policies on green economic efficiency and the mediating effect of innovation factor agglomeration.

According to the results obtained from the study: (1). during the inspection period, the impact of public participation landscaping on the green economy's efficiency tends to be "inverted U", while command control and market incentive regulating policies hinder the development of green economic efficiency; (2). The combination of innovative elements plays an important mediating role in the transmission of environmental regulation that affects green economic efficiency, but the mediating effects of different types of environmental regulation are somewhat different.

In the study, 4 different hypotheses are tested. These hypotheses are as follows:

H1: Command-controlled environmental regulation policies can inhibit the improvement of green economic efficiency.

H2: Market-incentive environmental regulation policies can inhibit the improvement of green economic efficiency.

H3: Public participation environmental regulation policies can promote the improvement of green economic efficiency.

H4: The agglomeration of innovation factors generated by environmental regulation policies can positively promote the improvement of green economic efficiency.

My opinions and suggestions regarding the study are as follows:

The topic of the article is current and interesting. I found the article generally successful. However, there are some typos in the study that need to be corrected. In addition, it would be appropriate to check the language of the article by a native speaker. Some sentences are very long so it is difficult to understand. These parts need to be simplified. The contributions of the study to the literature are clearly stated in the introduction. This part is sufficient. In addition, reference is made to the current literature. I think the literature review is sufficient.

It would be appropriate to give descriptive statistics of the data. If the abbreviations of the variables are included in this table, it will be easier for the reader to read. Necessary comments were made regarding the coefficients. Numerous control variables were used. Robustness tests have been carried out. The issue of endogeneity has been taken into account. For this reason, I think econometric estimations are suitable for the purpose of the article. In the conclusion part, policy imlications are included. This part is written in detail. However, the limitations of the study and what can be done in future studies were not mentioned. It would be appropriate to add these parts.

Author Response

(The authors gave the same response as above.)

Reviewer 3 Report

The paper provides an interesting perspective on “environmental regulation, innovation factor agglomeration and green economic efficiency”. The title and abstract are appropriate for the content of the text. Furthermore, the article is well constructed and organized, and analysis was well performed.

I just suggest authors to focus and highlight more clearly on the aim of this study.

What is the “spiritual demands of the public” (line 153)? I suggest authors to explain this concept.

The findings are well-written.

I appreciated the use of hypotheses; however, I would suggest the authors add a summary table of the expected hypotheses, in order to give an immediate overview of what is accepted or rejected, providing a brief discussion.

Elaborate more on the implications of the study to further research at the conclusion section of the study. 

Author Response

Please refer to the attachment "Reply to the review report"
